# Predictors of iron deficiency anaemia among children aged 6–59 months in Tanzania: Evidence from the 2015–16 TDHS-MIS cross-sectional household survey

Rose V. Msaki[1], Elizabeth Lyimo[1], Ray M. Masumo[1]*, Eliasaph Mwana[1], Doris Katana[1], Nyamizi Julius[1], Adeline Munuo[1], Germana Leyna[1], Abukari I. Issaka[2], Mansi V. Dhami[3], Kingsley E. Agho[3,4,5]

1 Tanzania Food and Nutrition Centre, Dar es Salaam, Tanzania, 2 School of Sciences, Western Sydney University, Penrith, NSW, Australia, 3 Translational Health Research Institute, School of Medicine, Western Sydney University, Campbelltown Campus, Penrith, NSW, Australia, 4 School of Health Sciences, Western Sydney University, Campbelltown Campus, Penrith, NSW, Australia, 5 African Vision Research Institute, University of KwaZulu-Natal, Durban, South Africa

* rmasumo@yahoo.com

**Data Availability Statement:** All datasets underlying this study are freely available at the DHS

## Abstract

Iron deficiency anaemia remains a public health problem, particularly in children aged 6–59 months. This study assessed factors associated with iron deficiency anaemia among children aged 6–23 months, 24–59 months and 6–59 months in Tanzania. Data for this cross-sectional study were extracted from the 2015–16 Tanzania Demographic and Health Survey and Malaria Indicator Survey (2015–16 TDHS-MIS). The study covered 8014 children aged 6–59 months and their mothers. Iron deficiency anaemia was defined (haemoglobin < 11g/dL). Univariable and multivariable logistic regression analyses that adjust for clustering and sampling weights were conducted to describe the associations between anaemia and potential confounding variables. The prevalence of iron deficiency anaemia among children aged 6–23 months, 24–59 months and 6–59 months were 76%, 49% and 59%, respectively. Factors associated with increased odds of iron deficiency anaemia among children aged 6–23 months included a mother being employed, being a male child, child perceived to small size at birth by mothers, a mother being anaemic and children belonging to the poorest socio-economic quintile. In addition, being a mother with no schooling, children not being dewormed, a mother being anaemic, delivering a baby at home, child fever and stunting, were factors associated with increased odds of iron deficiency anaemia among children aged 24–59 months. Factors associated with increased odds of iron deficiency anaemia among children aged 6–59 months were: a mother being employed, being a mother with no schooling, being a male child, belonging to the 6–59 months age bracket, a mother having a BMI of between 19 and 25 kg/m$^2$, a mother being anaemic, having a baby at home, children belonging to bigger households, child fever and stunting. Interventions to minimise the burden of iron deficiency anaemia in children should target employed and/or anaemic mothers, poor and rich households, as well as male children.

public repository. URL: https://dhsprogram.com/data/dataset/Tanzania_Standard-DHS_2015.cfm?flag=0.

**Funding:** The authors received no specific funding for this work.

**Competing interests:** The authors declare that they have no competing interests.

## Introduction

Anaemia, which is identified as a condition in which the blood haemoglobin concentration is lower than normal [1], and leads to poor cognitive and motor development in children and loss of work productivity in adulthood [2, 3], is a global health problem, particularly in developing countries [4, 5]. Iron deficiency anaemia has been found to be the major cause of anaemia, accounting for about 50% of all anaemia cases [3]. This proportion, however, may vary substantially across regions and countries [6]. In addition, anaemia could also be attributable to the effects of infectious diseases (particularly HIV, malaria, and helminth infections) [7] and genetic disorders of haemoglobin as well as other micronutrient deficiencies such as folic acid, vitamin A and vitamin B12 deficiencies [2].

As of 2015, it was estimated that 273 million (~ 42.6%) of under-5 children were anaemic, worldwide; whereas 60.2% of under-5 children in the African region were anaemic [8]. In Tanzania, anaemia among under-5 children remains high (58%) despite the efforts made by the government and other stakeholders in improving maternal, infant and young child health and nutrition services. Tanzanian children under 24 months of age are the most affected, with a peak prevalence of 81% among children aged 9–11 months [2, 9].

The situation of haemoglobinopathies causing hereditary haemolytic anaemia in Tanzania is very grim. A hospital-based, descriptive cross-sectional design was used to recruit newborns in 2009 and found that, out of 2,053 samples analysed, the prevalence of haemoglobinopathies was 18.2% (n = 374). The frequency of occurrence of abnormal haemoglobins was highest among participants whose parental origin was Coastal and Lake Zones [10]. Unfortunately, there is no much attempt at the national level to enlarge the epidemiological database and establishing specific programmes for screening populations at risk [11].

There has been a dearth of evidences on factors associated with Anaemia in children in Tanzania. For instance, Kejo and colleagues examined the prevalence of anaemia and its predictors among under-five children [12]. They used multivariable logistic regression to identify predictors of anaemia in children that include low birth weight, not consuming meat, not consuming vegetables, drinking milk and drinking. They concluded that low birth weight and dietary factors (i.e., low or non-consumption of iron-rich foods like meat, vegetables, and fruits) were predictors of anaemia among under-five children. However, the study by Kejo et al. [12] was limited in scope, as it covered only under-5 children in Arusha district. Another Tanzanian study by Simbauranga et al [13] sought to determine the prevalence and morphological types of anaemia and factors associated with severe anaemia in under-five children. In that study, factors related to child anaemia included unemployment of the parent, malaria parasitaemia and presence of sickle haemoglobin. However, this study was limited in scope too; as it covered only under-5 children admitted at Bugando Medical Centre in Mwanza district of Tanzania.

Due to the fact that anaemia is important in a national and global context; and given the limited studies on this burden in Tanzania [12–14], this current study utilized secondary data from the 2015–2016 Tanzania Demographic and Health Survey and Malaria Indicator Survey (TDHS-MIS) to assess the predictors of iron deficiency anaemia, the major cause of anaemia among children aged 6–23 months, 24–59 months and 6–59 months in Tanzania. The study results will support decision-makers, programme implementers, and academia at various planning, implementation, and training interventions focusing on prevention and control of nutrition anaemia.

### The theoretical framework of factors associated with iron deficiency anaemia in children

To analyse factors associated with iron deficiency anaemia in children in Tanzania, this current study adapted the Mosley and Chen conceptual framework to analyse child survival in

developing countries [15]. This framework, which was proposed in 1984 by Mosley and Chen, incorporated both social and biological factors associated with the survival of an infant. The adapted framework is presented in Fig 1.

## Methods

### Ethics statement

This study was based on an analysis of existing public domain survey datasets that are freely available online with all identifier information removed. The survey was approved by the Tanzania Ethics Committees i.e. the National Institute for Medical Research (NIMR) and, Zanzibar Medical Research and Ethics Committee (ZAMREC). Further, the survey was approved by Institutional Review Board of ICF International and the Centers for Disease Control and Prevention in Atlanta. Written consent was obtained from all respondents (parent/guardian) after being read a document emphasizing the voluntary nature of the survey and, all information was collected confidentially.

### Study design and study area

This was an analytical cross-sectional study conducted using nationally representative secondary data from the 2015–16 TDHS-MIS survey, which was conducted from August 2015 to March 2016 by the Tanzania National Bureau of Statistics (NBS) and the office of Chief Government Statistician (OCGS), Zanzibar, in collaboration with the Ministry of Health, Community Development, Gender, Elderly, and Children (MoHCDGEC) in Tanzania Mainland and the Ministry of Health, Zanzibar. The study included data both from urban and rural areas in the 30 regions of the United Republic of Tanzania, of which 25 regions were of Mainland Tanzania and 5 regions of Zanzibar [9]. The Tanzania DHS employs a multi-stage sampling procedure. The first stage involves a selection of a stratified sample from a list of enumeration areas (EAs) that have been obtained from the 2012 Population and Housing Census for United Republic of Tanzania conducted on the 26th August, 2012. These EAs are the clusters. This sample of EAs is selected with considerations to probability proportional to size (PPS) that takes into account the size of the enumeration area. At the second stage, after a complete list of households is available in each of the selected EAs, a fixed/variable number of households are selected by equal probability systematic sampling technique.

### Study population and sample size

The main objective of the 2015–16 TDHS-MIS survey was to provide up-to-date estimates of basic demographic and health indicators, including nutrition indicators for the prevalence of iron deficiency anaemia among children aged 6–59 months. The survey involved completed interviews with 13,266 ever-married women aged 15–49 years (response rate 97%) from sampled households. The survey utilized four questionnaires: a household, women's, men's and biomarker questionnaire. A household questionnaire was used to collect socio-demographic data for all household members, while the women questionnaire was used to gather information about maternal and child health. The Biomarker questionnaire was used to document anthropometric measurements (height and weight) and the iron deficiency anaemia test results for the children aged 6–59 months and the women of reproductive age 15–49 yrs. The present analysis was restricted to children aged 6–59 months and lived with the respondent (women aged 15–49 years). This yielded a weighted total of 8014 children aged 6–59 months.

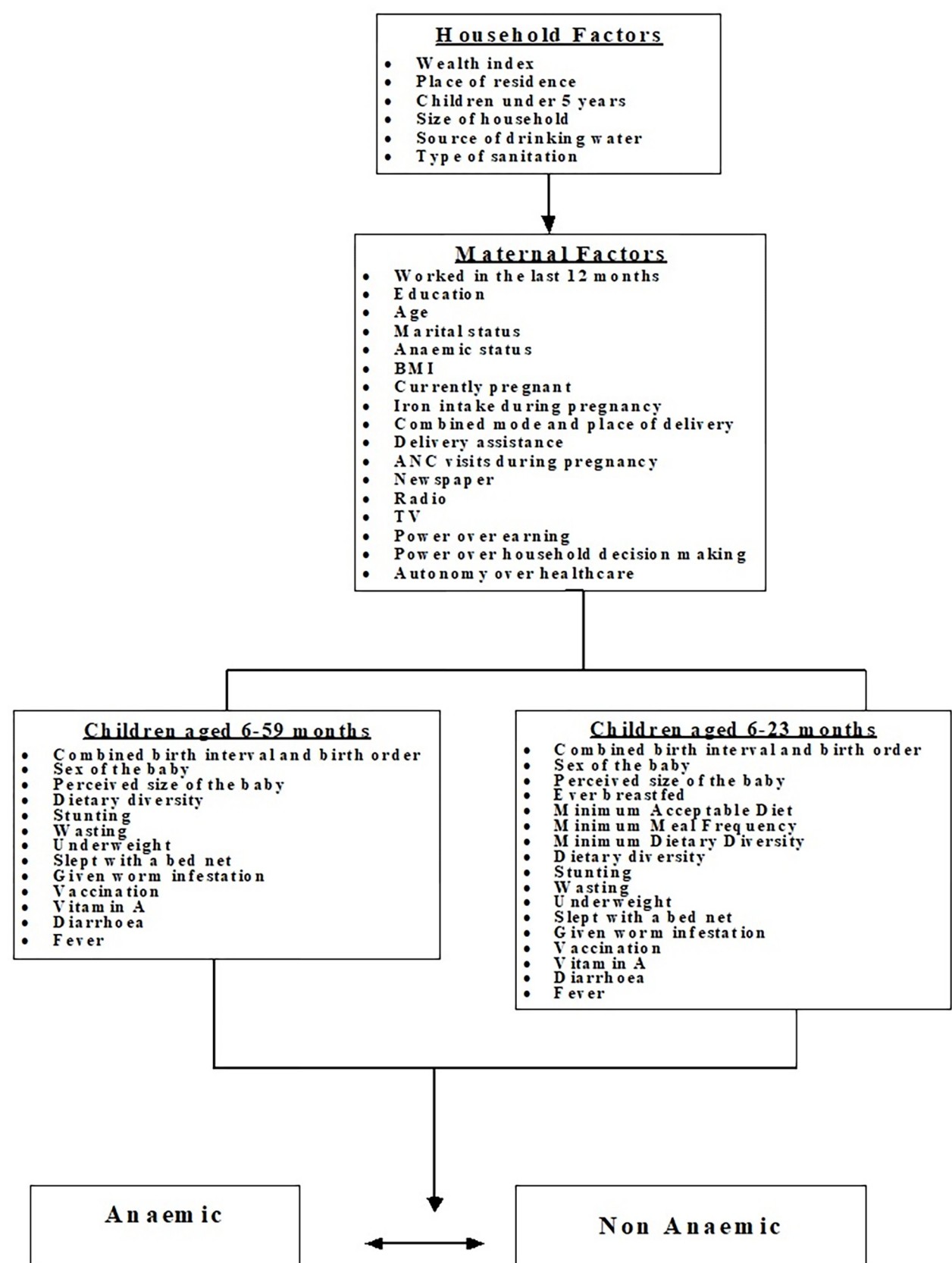

**Fig 1. Flow chart for factors associated with anaemia among children 6–59 months.**

## Study outcomes

The outcome variable for this analysis is iron deficiency anaemia among children aged 6–59 months. iron deficiency anaemia defined as a condition with a haemoglobin level below 11.0 g/dl as defined by the WHO [1].

## Predictor variables

The choice of independent variables for this study was informed by previous literature [16–19] and their availability in the 2015–16 TDHS dataset. These variables were categorised into maternal factors, child factors and household factors.

Participant's work status, level of education, age, marital status, anaemia status, body mass index (BMI), pregnancy status, uptake of iron during pregnancy, place of delivery, mode of delivery, type of delivery assistance, number of antenatal (ANC) clinic visits, access to newspaper/magazine, the radio and television, power over earnings, household decision making and autonomy over health care constituted the maternal factors. Child factors included: combined birth interval and birth order, sex, size at birth, age, breastfeeding status, meetings requirements for minimum acceptable diet (MAD), minimum meal frequency (MMF), minimum dietary diversity (MDD), MDD status, anthropometric status (stunting, wasting and underweight), use of bed net, uptake of a drug to prevent worm infestation, vaccination status and illness (diarrhoea and fever). The household factors were household wealth index, type of residence, number of under-5 children, household size, quality of the source of drinking water, and quality of toilet facility. The household wealth index was represented as a score of household assets through the principal components analysis method (PCA) [20]. Scores were assigned to each de jure household member after the index was computed, ranking each member of the sample by their score. In this study, the wealth index was categorized into five quintiles, namely, poorest, poorer, middle, rich and richest. The bottom 20% of the households were referred to as the poorest; the next bottom 20% was referred to as the poorer, the next bottom 20% was referred to as the middle-class, the next 20% was referred to as the richer, and the top 20% was referred to as the richest. Full details of the definition of the categorisation of all potential predictor variables used in the study are presented in Table 1.

## Anthropometric measurements

**Anthropometry.** Height and weight measurements were taken for all sampled children and women aged 15–49 years in Tanzania; including those not biological offspring of the women interviewed in the survey. Each trained interviewer carried a scale and measuring board. The scales were lightweight, bathroom-type with a digital screen. Recumbent heights were measured for children aged less than 24 months, whilst the standing heights of older children were measured [9].

**Testing for iron deficiency anaemia.** All the children aged 6–59 months in all selected households were tested for anaemia after receiving consent from their parents or guardians. Blood specimens were collected from all children aged 6–59 months and women of reproductive age 15–49 years for haemoglobin measurements. Blood samples were drawn from a drop of blood taken from a finger prick (or heel prick in children aged 6–11 months) and collected in the micro cuvette. Haemoglobin analysis was carried out on-site, using a battery-operated portable HemoCue analyser. Results for the anaemia test were given instantly to all the respondents. Parents/guardians of the children who had haemoglobin level less than 7 g/dl were advised to take the child to a health facility for treatment. The same advice was given to all women found to be anaemia, i.e. having a haemoglobin level (hb) of less than 7g/dl [1].

**Table 1. Definition and categorisation of variables used in the study.**

| Independent variables | Categorisation |
|---|---|
| *Household factors* | |
| Household wealth Index | hv217 (the household wealth index factor score) constructed by DHS based on a selected set of household assets. In quintiles, 1 = richest; 2 = richer; 3 = middle; 4 = poorer; 5 = poorest) |
| Residence | In the 2 following categories:(1 = urban; 2 = rural) |
| Number of children under 5 years | In the 3 following categories: (1 = None; 2 = 1 to 2; 3 = ≥3) |
| Household size | In the 2 following categories: (1 = 2–5; 2 = >6) |
| Source of drinking water | In the 2 following categories: (1 = improved; 2 = Unimproved) piped into dwelling, piped to yard or plot, piped to neighbour, public tap or standpipe, tube well or borehole, protected well, protected spring, tanker truck and bottled water. Unimproved otherwise |
| Type of toilet facility | In the 2 following categories: (1 = improved; 2 = Unimproved) Improved was flush to piped sewer system, flush to septic tank, flush to pit latrine, ventilated improved pit latrine, pit latrine with slab, and composting toilet. Unimproved otherwise. |
| *Maternal factors* | |
| Maternal working status | In the 2 following categories: (1 = not working; 2 = working (for the past 12 months)) |
| Maternal education | In the 3 following categories: (1 = Secondary or higher; 2 = Primary; 3 = No education) |
| Maternal age | In the 3 following categories: (1 = 15–19 years; 2 = 20–34 years; 3 = 35–49 years) |
| Marital status | In the 2 following categories: (1 = Currently married; 2 = divorced/ separated/widow) |
| Maternal anaemia | In the 2 following categories: (1 = Non anaemic; 2 = anaemic) haemoglobin (Hb) < 110g/dl for pregnancy women |
| Maternal BMI | In the 3 following categories: (1 = 25+ kg/m$^2$; 2 = 19–25 kg/m$^2$; 3 = ≤18.5kg/m$^2$) |
| Mothers currently pregnant | In the 2 following categories: (1 = Not at all; 2 = Yes) |
| Mothers taking iron during pregnancy | In the 2 following categories: (1 = Not at all; 2 = Yes) |
| Combined mode and place of delivery | In the 3 following categories: (1 = Vaginal; 2 = Caesarean; 3 = Home) |
| Type of delivery assistance | In the 2 following categories: (1 = Health professional[&]; 2 = non-Health professional) |
| Antenatal Clinic visits | In the 4 following categories: (1) 8+ antenatal care visits, (2) 4–7 antenatal care visits, and (3) 1–3 antenatal care visits (4) no antenatal care visits from a skilled provider for the most recent birth |
| mothers reading newspapers | In the 2 following categories: (1 = At least once a week; 2 = Less than once a week and 3 = Never) |
| mothers listening to radio | In the 2 following categories: (1 = At least once a week; 2 = Less than once a week and 3 = Never) |
| mothers watching television | In the 2 following categories: (1 = At least once a week; 2 = Less than once a week and 3 = Never) |
| Power over earning | In the 2 following categories:(1 = Yes; 2 = No) Yes- respondent alone, respondent and husband/partner. No- otherwise |
| Power over household decision making | In the 2 following categories:(1 = Yes; 2 = No) Yes- respondent alone, respondent and husband/partner. No- otherwise |
| Autonomy over health care | In the 2 following categories:(1 = Yes; 2 = No) Yes- respondent alone, respondent and husband/partner. No- otherwise |
| *Child Factors* | |

(*Continued*)

**Table 1.** (Continued)

| Independent variables | Categorisation |
|---|---|
| Combined birth interval and birth order | (1 = 1st birth rank; 2 = 2nd/3rd birth rank, more than 2 years interval; 3 = 2nd/3rd birth rank, less than or equal to 2 years; 4 = 4th birth rank, more than 2 years interval; 5 = 4th birth rank, less than or equal to 2 years) |
| Sex of baby | In the 2 following categories: (1 = Male; 2 = Female) |
| Size of baby | In the 3 following categories: (1 = Large; 2 = Average; 3 = Small) |
| Age of child (months) | In the 2 following categories: (1 = 6–23 months; 2 = 24–59 months.) |
| Child ever breastfed | In the 2 following categories: (1 = Never Breastfed; 2 = Ever Breastfed) |
| Minimum Acceptable Diet | In the 2 following categories: (1 = No; 2 = Yes/some) |
| Minimum Meal Frequency | In the 2 following categories: (1 = No; 2 = Yes/some) |
| Minimum Dietary Diversity | In the 2 following categories: (Yes; 1 = ($\geq$ 4 of 7 food groups) = No; 2 = ($<$ 4 of 7 food groups). |
| Dietary Diversity | In the 3 following categories: (1 = High Diversity; 2 = Moderate Diversity; 3 = Low Diversity) High diversity (5–7 food groups); moderate diversity (3–4 food groups) & low diversity (0–2 food groups) |
| Stunting | In the 2 following categories: (1 = Not stunted; 2 = stunted (height-for-age z-score $<$ -2 SD) |
| Wasting | In the 2 following categories: (1 = Not wasted; 2 = wasted (weight-for-height z-score $<$ -2 SD) |
| Underweight | In the 2 following categories: (1 = Not underweight; 2 = underweight (weight-for-age z-score $<$ -2 SD) |
| Children under 5 years who slept with bed net | In the 2 following categories: (1 = No; 2 = Yes) |
| Children given drugs for preventing worm infestation | In the 2 following categories: (1 = No; 2 = Yes) |
| Children fully vaccinated | In the 2 following categories: (1 = None; 2 = full) |
| Vitamin A | In the 2 following categories: (1 = No; 2 = Yes) |
| Had diarrhoea recently | In the 2 following categories: (1 = No; 2 = Yes) |
| Had fever in last 2 weeks | In the 2 following categories: (1 = No; 2 = Yes) |

## Statistical analysis

All analyses were conducted using Stata version 14.0 (Stata Corp, College Station, Texas, USA) to determine factors associated with iron deficiency anaemia among children aged 6–23 months, 24–59 months and 6–59 months in Tanzania, the dependent variable was expressed as a dichotomous one; that is, category 0 (not anaemic (hb $>$ 11g/dl) and category 1 (anaemic (hb $<$ 11g/dl). Preliminary analyses involved the frequency distribution of all potential predictor variables used in this study. An estimation of the prevalence and corresponding confidence intervals of iron deficiency anaemia among children aged 6–23 months, 24–59 months and 6–59 months then followed. We then performed univariate binary logistic regression analysis that adjusts for clustering and sampling weights to examine the association between anaemic children aged 6–23 months and overall anaemia among children aged 6–59 months and multivariable survey logistic regression analyses that adjust for clustering and sampling weights to examine the factors associated with anaemia after controlling for confounding variables (Household-, maternal- and child-level factors).

A manual process elimination method was used in the multivariable analysis models to identify factors that were significantly associated with the study outcome, using a 5% significance level. In order to avoid or minimise any statistical errors in our analysis, we repeated the manual elimination process by using a different approach. This involved three steps: (1) only potential risk factors with P-value $<$ 0.20 were entered in the backward elimination process,

(2) the backward elimination was tested by including all variables (all potential risk factors); and, (3) Any collinearity was tested and reported in the final model. The odds ratios with 95% confidence intervals (CIs) were calculated in order to assess the adjusted risk of independent variables; and those with P < 0.05 were retained in the final model.

## Results

### Characteristics of the sample

Of the 8014 mothers (of children aged 6–59 months) considered in this study, 73.1% were unemployed (Table 2). Approximately one out of every five mothers (21.1%) had no schooling, and the majority of the mothers were aged 20–34 years (68.8%) and never been married (87.4%). Less than half of the mothers (48.3%) were anaemic, and only about a quarter of them (25.1%) had a BMI of above 25 kg/m$^2$. Only 10.4% of the mothers were pregnant, and out of these, the majority (80.3%) took iron during their pregnancy. The majority of the mothers (62.5%) delivered their babies at a health facility, and approximately nine out of ten of them (94.2%) were delivered through a non-caesarean section. More than 60% of the mothers were delivered with the help of health professionals, and more than one-third of them (34.1%) did not attend any ANC clinics. The proportion of mothers who had no access to newspapers/magazines, the radio and television was 63.8%, 25.4% and 57.7%, respectively. The proportion of mothers alone having power over earnings and household decision making as well as autonomy over health care was 46.7%, 66.8% and 58.7%, respectively. For children aged 6–59 months, close to a quarter (24.5%) of them were firstborn. The proportion of males, small size and children aged 24–59 months was 50.6%, 10.2% and 64.1%, respectively. The proportion of children aged 6–23 months who were ever breastfed and met the requirements for MAD, MMF and MDD, and being exposed to low dietary diversity was 97.9%, 6.8%, 33.8%, and 22.9%, respectively. A large majority (85.6%) of children aged 6–59 months were exposed to low dietary diversity. The proportion of children aged 6–59 months who were stunted, wasted and underweight was 36.6%, 4.3% and 14.4%, respectively. The proportion of children aged less than 5 years who did not sleep under bed net, children aged 6–59 months who were not given any drugs to prevent worm infestation, were not fully vaccinated, did not take vitamin A, had no diarrhoea and had no fever was 33.6%, 62.6%, 99%, 58.5%, 87.1% and 80.7% respectively.

Close to a quarter (24%) of children aged 6–59 months lived in the poorest households and approximately 73% of them lived in rural areas. More than half the children aged 6–59 months lived in households with more than six people (59.7%), improved source of drinking water (54.4%) and unimproved toilet facility (55.9%).

As shown in Fig 2, the prevalence of iron deficiency anaemia among children aged 6–23 months and 24–59 months were 75.8% and 48.6%, respectively. However, the figure showed that about two-thirds of the children aged 6–59 months (58.8%) are anaemic.

### Factors associated with iron deficiency anaemia in children aged 6–23 months

Factors significantly associated with increased odds of anaemia among children aged 6–23 months included: being a working mother [Adjusted odds ratio (OR): 1.43; 95% confidence interval (CI): (1.13, 1.82)], being a male child [OR: 1.46; 95% CI: (1.17, 1.82)], child being considered small at birth [OR: 1.77; 95% CI: (1.22, 2.56)], being an anaemic mother [OR: 2.28; 95% CI: (1.87, 2.79)] and a child belonging to the poorest socioeconomic quintile [OR: 2.28; 95% CI: (1.87, 2.79)] (Table 3).

**Table 2. Household, maternal, and child-level characteristics of children aged 6–23 months, 24–59 months and 6–59 months, Tanzania, 2015–16 (n = 8014).**

| Characteristics | 6–23 months | | 24–59 months | | 6–59 months | |
|---|---|---|---|---|---|---|
| | n (weighted) | Percentage (%) | n (weighted) | Percentage (%) | n (weighted) | Percentage (%) |
| *Household characteristics* | | | | | | |
| *Household Wealth Index* | | | | | | |
| Richest | 562 | 17.4 | 961 | 16.7 | 1524 | 17.0 |
| Richer | 614 | 19.0 | 1055 | 18.3 | 1669 | 18.6 |
| Middle | 605 | 18.8 | 1119 | 19.4 | 1725 | 19.2 |
| Poorer | 682 | 21.1 | 1229 | 21.3 | 1911 | 21.3 |
| Poorest | 762 | 23.6 | 1394 | 24.2 | 2156 | 24.0 |
| *Place of residence* | | | | | | |
| Urban | 910 | 28.2 | 1534 | 26.6 | 2444 | 27.2 |
| Rural | 2315 | 71.8 | 4225 | 73.4 | 6540 | 72.8 |
| *Number of children under 5 years* | | | | | | |
| None | 144 | 5.3 | 319 | 6.5 | 463 | 6.1 |
| 1 to 2 | 2301 | 84.4 | 4109 | 84.0 | 6409 | 84.2 |
| ≥3 | 280 | 10.3 | 464 | 9.5 | 744 | 9.8 |
| *Household Size (grouped)* | | | | | | |
| 1 to 5 | 1311 | 40.6 | 2351 | 40.8 | 3662 | 40.8 |
| >6 | 1915 | 59.4 | 3407 | 59.2 | 5322 | 59.2 |
| *Source of drinking water* | | | | | | |
| Not improved | 1441 | 44.7 | 2652 | 46.1 | 4093 | 45.6 |
| Improved | 1785 | 55.3 | 3106 | 53.9 | 4891 | 54.4 |
| *Type of toilet facility* | | | | | | |
| Improved | 862 | 44.6 | 1512 | 43.8 | 2374 | 44.1 |
| Unimproved | 1072 | 55.4 | 1942 | 56.2 | 3014 | 55.9 |
| *Maternal characteristics* | | | | | | |
| *Respondent worked in the last 12 months* | | | | | | |
| Non-working | 2257 | 70.0 | 4311 | 74.9 | 6568 | 73.1 |
| Working | 968 | 30.0 | 1448 | 25.1 | 2416 | 26.9 |
| *Mother's Education* | | | | | | |
| Secondary | 561 | 17.4 | 713 | 12.4 | 1274 | 14.2 |
| Primary | 2044 | 63.4 | 3771 | 65.5 | 5815 | 64.7 |
| No education | 621 | 19.3 | 1274 | 22.1 | 1895 | 21.1 |
| *Mother's age (years)* | | | | | | |
| 15–19 | 358 | 11.1 | 149 | 2.6 | 507 | 5.6 |
| 20–34 | 2247 | 69.7 | 3930 | 68.3 | 6177 | 68.8 |
| 35–49 | 621 | 19.3 | 1679 | 29.2 | 2300 | 25.6 |
| *Mother's marital status* | | | | | | |
| Never in marriage | 2618 | 88.6 | 4820 | 86.7 | 7438 | 87.4 |
| Currently married | 335 | 11.3 | 741 | 13.3 | 1077 | 12.6 |
| *Maternal anaemia* | | | | | | |
| Non-anaemic | 1684 | 54.0 | 2803 | 50.5 | 4487 | 51.7 |
| Anaemic | 1437 | 46.0 | 2751 | 49.5 | 4188 | 48.3 |
| *Maternal BMI (Kg/m2)* | | | | | | |
| 25+ | 695 | 21.8 | 1559 | 27.2 | 2254 | 25.3 |
| 18.5–24.9 | 2336 | 73.1 | 3922 | 68.5 | 6258 | 70.2 |
| <18.5 | 163 | 5.1 | 243 | 4.2 | 406 | 4.5 |
| *Currently Pregnant* | | | | | | |

*(Continued)*

**Table 2.** (Continued)

| Characteristics | 6–23 months | | 24–59 months | | 6–59 months | |
|---|---|---|---|---|---|---|
| | n (weighted) | Percentage (%) | n (weighted) | Percentage (%) | n (weighted) | Percentage (%) |
| No | 3013 | 93.4 | 5038 | 87.5 | 8051 | 89.6 |
| Yes | 213 | 6.6 | 720 | 12.5 | 933 | 10.4 |
| *Taking iron during pregnancy* | | | | | | |
| No | 567 | 18.3 | 628 | 21.1 | 1195 | 19.7 |
| Yes | 2528 | 81.7 | 2350 | 78.9 | 4878 | 80.3 |
| *Combined mode and place of delivery* | | | | | | |
| Vaginal | 1887 | 58.4 | 3203 | 55.6 | 56.65 | 56.7 |
| Caesarean section | 216 | 6.7 | 306 | 5.3 | 5.809 | 5.8 |
| Home | 1123 | 34.8 | 2250 | 39.1 | 37.54 | 37.5 |
| *Delivery assistance* | | | | | | |
| Health professional | 2083 | 64.5 | 3533 | 61.4 | 5616 | 62.5 |
| Non-health professional | 1143 | 35.4 | 2226 | 38.7 | 3368 | 37.5 |
| *Antenatal Clinical visits* | | | | | | |
| 8+ | 29 | 0.9 | 74 | 1.3 | 103 | 1.2 |
| 4 to 7 | 1527 | 47.2 | 1576 | 27.3 | 3103 | 34.5 |
| 1 to 3 | 1465 | 45.4 | 1282 | 22.3 | 2747 | 30.6 |
| None | 208 | 6.5 | 2855 | 49.1 | 3063 | 33.7 |
| *Mother read magazine or newspaper* | | | | | | |
| At least | 320 | 9.9 | 506 | 8.8 | 826 | 9.2 |
| Less than | 874 | 27.1 | 1556 | 27.0 | 2430 | 27.1 |
| Never | 2031 | 63.0 | 3695 | 64.2 | 5726 | 63.8 |
| *Mother listened to radio* | | | | | | |
| At least | 1335 | 41.4 | 2272 | 39.5 | 3607 | 40.2 |
| Less than | 1092 | 33.9 | 2000 | 34.7 | 3092 | 34.4 |
| Never | 799 | 24.8 | 1486 | 25.8 | 2285 | 25.4 |
| *Mother watched television* | | | | | | |
| At least | 630 | 19.5 | 1018 | 17.7 | 1648 | 18.3 |
| Less than | 764 | 23.7 | 1390 | 24.1 | 2154 | 24.0 |
| Never | 1831 | 56.8 | 3351 | 58.2 | 5182 | 57.7 |
| *Mother Power over earnings* | | | | | | |
| No | 1766 | 54.8 | 3027 | 52.6 | 4793 | 53.4 |
| Yes | 1459 | 45.2 | 2731 | 47.4 | 4191 | 46.7 |
| *Mother Power over household decision making* | | | | | | |
| No | 1096 | 34.0 | 1891 | 32.8 | 2987 | 33.3 |
| Yes | 2129 | 66.0 | 3868 | 67.2 | 5997 | 66.8 |
| *Mother autonomy over health care* | | | | | | |
| No | 1345 | 41.7 | 2362 | 41.0 | 3707 | 41.3 |
| Yes | 1880 | 58.3 | 3396 | 59.0 | 5277 | 58.7 |
| *Child characteristics* | | | | | | |
| *Combined birth interval and birth order* | | | | | | |
| 1st birth rank | 875 | 27.1 | 1323 | 23.0 | 2,198 | 24.5 |
| 2nd/3rd birth rank, more than 2 years interval | 1140 | 35.4 | 2090 | 36.3 | 3230 | 36.0 |
| 2nd/3rd birth rank, less than or equal to 2 | 308 | 9.5 | 594 | 10.3 | 902 | 10.0 |
| 4th birth rank, more than 2 years interval | 704 | 21.8 | 1336 | 23.2 | 2040 | 22.7 |
| 4th birth rank, less than or equal to 2 | 198 | 6.1 | 416 | 7.2 | 614 | 6.8 |
| *Sex of child* | | | | | | |

(Continued)

**Table 2.** (Continued)

| Characteristics | 6–23 months | | 24–59 months | | 6–59 months | |
|---|---|---|---|---|---|---|
| | n (weighted) | Percentage (%) | n (weighted) | Percentage (%) | n (weighted) | Percentage (%) |
| Female | 1588 | 49.2 | 2849 | 49.5 | 4437 | 49.4 |
| Male | 1637 | 50.8 | 2909 | 50.5 | 4547 | 50.6 |
| *Perceived size of baby* | | | | | | |
| Large | 619 | 19.4 | 1101 | 19.3 | 1720 | 19.3 |
| Average | 2224 | 69.7 | 4060 | 71.0 | 6284 | 70.5 |
| Small | 350 | 11.0 | 554 | 9.7 | 904 | 10.2 |
| *Child's age (months)* | | | | | | |
| 6 to 23 | | | | | 3225 | 35.9 |
| 24 to 59 | | | | | 5758 | 64.1 |
| *Ever Breastfed* | | | | | | |
| No | 53 | 2.1 | | | | |
| Yes | 2444 | 97.9 | | | | |
| *Minimum Acceptable Diet* | | | | | | |
| No | 3005 | 93.2 | | | | |
| Yes | 220 | 6.8 | | | | |
| *Minimum meal frequency* | | | | | | |
| No | 2134 | 66.2 | | | | |
| Yes | 1091 | 33.8 | | | | |
| *Minimum Dietary Diversity* | | | | | | |
| No | 2488 | 77.1 | | | | |
| yes | 738 | 22.9 | | | | |
| *Dietary Diversity* | | | | | | |
| High | 236 | 10.1 | 105 | 2.0 | 340 | 4.5 |
| Moderate | 502 | 21.5 | 251 | 4.8 | 753 | 9.9 |
| Low | 1602 | 68.5 | 4917 | 93.3 | 6519 | 85.6 |
| *Stunted* | | | | | | |
| No | 2047 | 68.5 | 2912 | 60.3 | 4959 | 63.4 |
| Yes | 940 | 31.5 | 1919 | 39.7 | 2859 | 36.6 |
| *Wasted* | | | | | | |
| No | 2809 | 94.0 | 4661 | 96.8 | 7470 | 95.7 |
| Yes | 179 | 6.0 | 153 | 3.2 | 332 | 4.3 |
| *Underweight* | | | | | | |
| No | 2593 | 86.4 | 4125 | 85.2 | 6718 | 85.6 |
| Yes | 410 | 13.7 | 718 | 14.8 | 1128 | 14.4 |
| *Children <5 slept with bed net* | | | | | | |
| Yes | 2150 | 68.8 | 3563 | 65.0 | 5713 | 66.4 |
| No | 975 | 31.2 | 1918 | 35.0 | 2893 | 33.6 |
| *Given drugs for preventing worm infestation* | | | | | | |
| No | 2190 | 71.1 | 3124 | 57.8 | 5314 | 62.6 |
| Yes | 891 | 28.9 | 2278 | 42.2 | 3170 | 37.4 |
| *Fully Vaccinated* | | | | | | |
| No | 3208 | 99.5 | 5690 | 98.8 | 8897 | 99.0 |
| Yes | 18 | 0.6 | 69 | 1.2 | 87 | 1.0 |
| *Vitamin A* | | | | | | |
| No | 1739 | 56.9 | 3114 | 59.4 | 4853 | 58.5 |
| Yes | 1320 | 43.2 | 2126 | 40.6 | 3446 | 41.5 |

*(Continued)*

**Table 2.** (Continued)

| Characteristics | 6–23 months | | 24–59 months | | 6–59 months | |
|---|---|---|---|---|---|---|
| | n (weighted) | Percentage (%) | n (weighted) | Percentage (%) | n (weighted) | Percentage (%) |
| *Had diarrhoea in the last two weeks* | | | | | | |
| No | 2402 | 78.6 | 4780 | 92.2 | 7182 | 87.1 |
| Yes | 654 | 21.4 | 406 | 7.8 | 1061 | 12.9 |
| *Had fever in the last two weeks* | | | | | | |
| No | 2376 | 77.7 | 4316 | 82.5 | 6692 | 80.7 |
| yes | 682 | 22.3 | 914 | 17.5 | 1596 | 19.3 |

## Factors associated with iron deficiency anaemia in children aged 24–59 months

Being a mother with no schooling [OR: 1.54; 95% CI: (1.18, 2.00)], being an anaemic mother [OR: 1.77; 95% CI: (1.54, 2.04)], having a baby delivered at home [OR: 1.27; 95% CI: (1.09, 1.48)], a child having fever [OR: 1.61; 95% CI: (1.32, 1.95)] and a child being stunted [OR: 1.46; 95% CI: (1.25, 1.71)] were the factors associated with increased odds of anaemia among

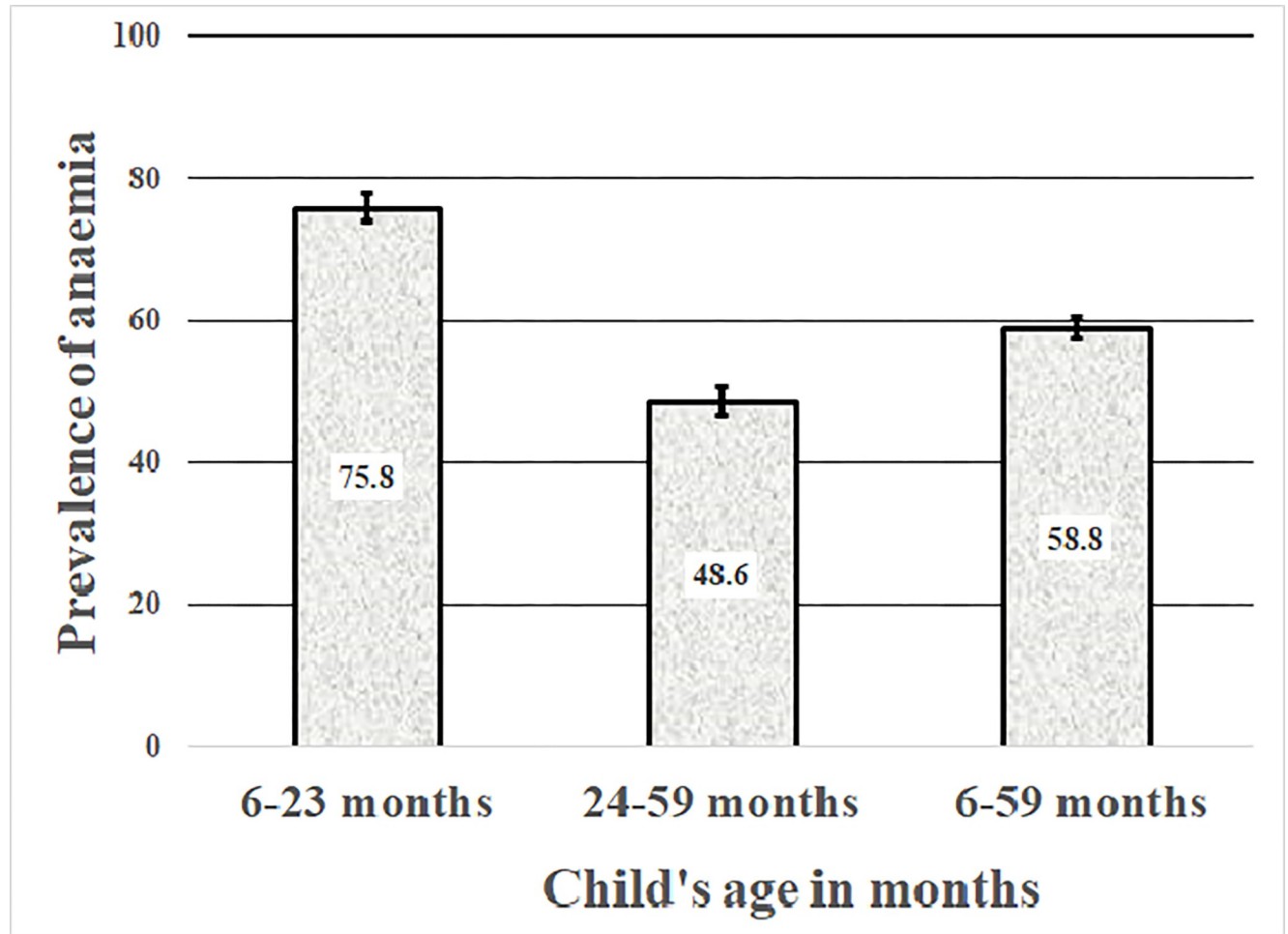

**Fig 2. Prevalence and 95% confidence intervals of anaemia among children aged 6–23 months, 24–59 months and 6–59 months.**

**Table 3. Univariate and multivariate analyses of the odds of a child aged 6–23 months being anaemic–Tanzania 2015–16.**

| Characteristics | Unadjusted | | Adjusted | |
|---|---|---|---|---|
| | OR (95% CI) | P-Value | OR (95% CI) | P-Value |
| *Household Factors* | | | | |
| **Household Wealth Index** | | | | |
| Richest | 1.00 | | 1.00 | |
| Richer | 1.13 (0.83,1.53) | 0.445 | 1.04 (0.75, 1.45) | 0.791 |
| Middle | 1.24 (0.91,1.71) | 0.172 | 1.29 (0.91, 1.82) | 0.142 |
| Poorer | 1.18 (0.86,1.61) | 0.303 | 1.20 (0.85, 1.69) | 0.279 |
| Poorest | 1.44 (1.08,1.94) | 0.014 | 1.50 (1.10, 2.05) | 0.010 |
| **Type of residence** | | | | |
| Urban | 1.00 | | | |
| Rural | 1.12 (0.91,1.39) | 0.281 | | |
| **Number of children under 5 years** | | | | |
| None | 1.00 | | | |
| 1 to 2 | 0.86 (0.36,2.09) | 0.744 | | |
| ≥3 | 1.43 (0.53,3.84) | 0.476 | | |
| **Household Size (grouped)** | | | | |
| 1 to 5 | 1.00 | | | |
| >6 | 1.23 (1.01,1.51) | 0.043 | | |
| **Source of drinking water** | | | | |
| Not improved | 1.00 | | | |
| Improved | 0.78 (0.63,0.98) | 0.033 | | |
| **Type of toilet facility** | | | | |
| Improved | 1.00 | | | |
| Unimproved | 1.32 (1.02,1.72) | 0.036 | | |
| *Maternal Factors* | | | | |
| **Respondent worked in the last 12 months** | | | | |
| Non-working | 1.00 | | 1.00 | |
| Working | 1.47 (1.17,1.84) | 0.001 | 1.43 (1.13, 1.82) | 0.003 |
| **Mother's Education** | | | | |
| Secondary | 1.00 | | | |
| Primary | 1.08 (0.85,1.39) | 0.523 | | |
| No education | 1.43 (1.01,2.02) | 0.044 | | |
| **Mother's age (years)** | | | | |
| 15–19 | 1.00 | | | |
| 20–34 | 0.73 (0.51,1.03) | 0.075 | | |
| 35–49 | 0.64 (0.42,0.98) | 0.038 | | |
| **Mother's marital status** | | | | |
| Never in marriage | 1.00 | | | |
| Currently married | 0.99 (0.70,1.42) | 0.973 | | |
| **Maternal anaemia** | | | | |
| Non-anaemic | 1.00 | | 1.00 | |
| Anaemic | 1.44 (1.08,1.94) | <0.001 | 2.28 (1.87, 2.79) | <0.001 |
| **Maternal BMI (Kg/m$^2$)** | | | | |
| 25+ | 1.00 | | | |
| 18.5–24.9 | 1.34 (1.05,1.70) | 0.019 | | |
| <18.5 | 1.61 (1.01,2.56) | 0.045 | | |
| **Currently Pregnant** | | | | |

*(Continued)*

**Table 3.** (Continued)

| Characteristics | Unadjusted | | Adjusted | |
|---|---|---|---|---|
| | OR (95% CI) | P-Value | OR (95% CI) | P-Value |
| No | 1.00 | | | |
| Yes | 1.54 (0.97,2.44) | 0.065 | | |
| **Taking iron during pregnancy** | | | | |
| No | 1.00 | | | |
| Yes | 1.04 (0.80,1.37) | 0.754 | | |
| **Combined mode and place of delivery** | | | | |
| Vaginal | 1.00 | | | |
| Caesarean section | 0.68 (0.46,1.01) | 0.055 | | |
| Home | 1.05 (0.84,1.32) | 0.661 | | |
| **Delivery assistance** | | | | |
| Health professional | 1.00 | | | |
| Non-health professional | 1.01 (0.80,1.26) | 0.953 | | |
| **Antenatal Clinical visits** | | | | |
| 8+ | 1.00 | | | |
| 4 to 7 | 1.11 (0.34,3.51) | 0.861 | | |
| 1 to 3 | 1.21 (0.38,3.83) | 0.740 | | |
| None | 1.82 (0.53,6.39) | 0.345 | | |
| **Mother read magazine or newspaper** | | | | |
| At least | 1.00 | | | |
| Less than | 1.10 (0.77,1.57) | 0.612 | | |
| Never | 1.11 (0.79,1.55) | 0.554 | | |
| **Mother listened to radio** | | | | |
| At least | 1.00 | | | |
| Less than | 1.05 (0.85,1.31) | 0.650 | | |
| Never | 1.23 (0.95,1.58) | 0.113 | | |
| **Mother watched television** | | | | |
| At least | 1.00 | | | |
| Less than | 1.43 (1.06,1.94) | 0.020 | | |
| Never | 1.16 (0.91,1.47) | 0.239 | | |
| **Mother power over earnings** | | | | |
| No | 1.00 | | | |
| Yes | 0.90 (0.74,1.10) | 0.292 | | |
| **Mother power over household decision making** | | | | |
| No | 1.00 | | | |
| Yes | 0.82 (0.66,1.03) | 0.082 | | |
| **Mother autonomy over health care** | | | | |
| No | 1.00 | | | |
| Yes | 0.81 (0.66,0.99) | 0.038 | | |
| *Child Factors* | | | | |
| **Combined birth interval and birth order** | | | | |
| 1st birth rank | 1.00 | | 1.00 | |
| 2nd/3rd birth rank, more than 2 years interval | 0.80 (0.61,1.05) | 0.103 | 0.79 (0.60,1.04) | 0.103 |
| 2nd/3rd birth rank, less than or equal to 2 | 0.87 (0.61,1.25) | 0.458 | 0.87 (0.61,1.24) | 0.458 |
| 4th birth rank, more than 2 years interval | 0.91 (0.68,1.23) | 0.546 | 0.91 (0.67,1.22) | 0.546 |
| 4th birth rank, less than or equal to 2 | 0.74 (0.48,1.14) | 0.171 | 0.73 (0.47,1.14) | 0.171 |
| **Sex of child** | | | | |

(*Continued*)

**Table 3.** (Continued)

| Characteristics | Unadjusted | | Adjusted | |
|---|---|---|---|---|
| | OR (95% CI) | P-Value | OR (95% CI) | P-Value |
| Female | 1.00 | | 1.00 | |
| Male | 1.35 (1.10,1.67) | 0.004 | 1.46 (1.17, 1.82) | 0.001 |
| **Size of baby** | | | | |
| Large | 1.00 | | 1.00 | |
| Average | 1.36 (1.07,1.72) | 0.012 | 1.35 (1.06, 1.73) | 0.015 |
| Small | 1.79 (1.27,2.51) | 0.001 | 1.77 (1.22, 2.56) | 0.002 |
| **Ever Breastfed** | | | | |
| No | 1.00 | | | |
| Yes | 1.08 (0.25,4.66) | 0.917 | | |
| **Minimum Acceptable Diet** | | | | |
| No | 1.00 | | | |
| Yes | 0.93 (0.66,1.31) | 0.683 | | |
| **Minimum meal frequency** | | | | |
| No | 1.00 | | | |
| Yes | 1.16 (0.95,1.42) | 0.126 | | |
| **Minimum Dietary Diversity** | | | | |
| No | 1.00 | | | |
| Yes | 0.81 (0.64,1.02) | 0.083 | | |
| **Dietary Diversity** | | | | |
| High | 1.00 | | | |
| Moderate | 1.27 (0.82, 1.97) | 0.290 | | |
| Low | 1.47 (0.99, 2.14) | 0.051 | | |
| **Stunted** | | | | |
| No | 1.00 | | | |
| Yes | 1.06 (0.87,1.30) | 0.511 | | |
| **Wasted** | | | | |
| No | 1.00 | | | |
| Yes | 1.66 (1.01,2.70) | 0.042 | | |
| **Underweight** | | | | |
| No | 1.00 | | | |
| Yes | 1.32 (0.98,1.77) | 0.060 | | |
| **Children <5 slept with bed net** | | | | |
| Yes | 1.00 | | | |
| No | 1.04 (0.84,1.28) | 0.714 | | |
| **Given drugs for preventing worm infestation** | | | | |
| No | 1.00 | | | |
| Yes | 0.86 (0.69,1.05) | 0.157 | | |
| **Fully Vaccinated** | | | | |
| No | 1.00 | | | |
| Yes | 0.56 (0.14,2.28) | 0.426 | | |
| **Vitamin A** | | | | |
| No | 1.00 | | | |
| Yes | 0.79 (0.64,0.98) | 0.036 | | |
| **Had diarrhoea in the last two weeks** | | | | |
| No | 1.00 | | | |
| Yes | 1.20 (0.94,1.52) | 0.133 | | |

(*Continued*)

**Table 3.** (Continued)

| Characteristics | Unadjusted | | Adjusted | |
|---|---|---|---|---|
| | OR (95% CI) | P-Value | OR (95% CI) | P-Value |
| **Had fever in the last two weeks** | | | | |
| No | 1.00 | | | |
| Yes | 1.21 (0.94,1.55) | 0.128 | | |

children aged 24–59 months. The odds of being anaemic was significantly lower among children aged 24–59 months who were dewormed compared with those who were not [(OR: 0.84; 95% CI: 0.72, 0.98)] (Table 4).

## Factors associated with iron deficiency anaemia in children aged 6–59 months

Factors significantly associated with increased odds of anaemia among children aged 6–59 months were: being a working mother [OR: 1.2; 95% CI: (1.03, 1.38)], being a mother with no schooling [OR: 1.46; 95% CI: (1.17, 1.84)], being a male child [OR: 1.14; 95% CI: (1.02, 1.27)], being a child within the 6–59 months age bracket [OR: 3.66; 95% CI: 3.27, 4.1)], being a mother with BMI between 19 and 25 kg/m$^2$ [OR: 1.25; 95% CI: 1.08, 1.45)], being an anaemic mother [OR: 1.86; 95% CI: 1.65, 2.09)], having a baby at a non-institutional facility [OR: 1.15; 95% CI: (1.01, 1.32)], belonging to richer households [OR: 1.15; 95% CI: 1.02, 1.3)], a child having fever [OR: 1.44; 95% CI: 1.23, 1.68)], a child being stunted [OR: 1.31; 95% CI: 1.14, 1.5)], shown in Table 5.

## Discussion

This current study examined the prevalence and significant factors associated with iron deficiency anaemia among children aged 6 to 59 months in Tanzania. Our study revealed that the prevalence of iron deficiency anaemia was 75.8% and 48.6% for children aged 6–23 months and 24–59 months, respectively. Furthermore, 58.8% of children aged 6–59 months were found to be anaemic. Predictors of iron deficiency anaemia among children aged 6–23 months included: maternal work and anaemia status, sex and size and household wealth index of the child. Factors associated with iron deficiency anaemia among children aged 24–59 months included: maternal anaemia status, place of delivery of baby, illness and anthropometry. For children aged 6–59 months, factors associated with iron deficiency anaemia included: maternal work status, level of education, BMI, anaemia status, place of delivery, sex of child, size of household, illness and child anthropometry. The high prevalence of iron deficiency anaemia among children in the lowest age bracket (6–23 months) is higher than the average African prevalence of this condition of 60.2% [8]. The high prevalence of iron deficiency anaemia among children in the 6–23 month age bracket is consistent with that of a study from Uganda [21], which showed that child age (6 to 23 months) was significantly associated with anaemia. The Uganda study also revealed that the prevalence of iron deficiency anaemia was highest among the youngest age groups and generally reduced with an increase in the age of the children, similar to results from other past studies [22, 23]. This observation could be explained by the fact that iron stores are generally depleted among children by six months of age while the blood volume doubles from 4 to 12 months after birth. In this way, the dietary sources of iron are very important in keeping up with this rapid rate of red blood cell synthesis, which may give rise to anaemia if the dietary sources are inadequate [4, 5].

**Table 4. Univariate and multivariate analyses of the odds of a child aged 24–59 months being anaemic–Tanzania 2015–16.**

| Characteristics | Unadjusted | | Adjusted | |
|---|---|---|---|---|
| | OR (95% CI) | P-Value | OR (95% CI) | P-Value |
| *Household Factors* | | | | |
| **Household Wealth Index** | | | | |
| Richest | 1.00 | | | |
| Richer | 0.98 (0.76,1.28) | 0.910 | | |
| Middle | 1.54 (1.18,2.01) | <0.001 | | |
| Poorer | 1.73 (1.34,2.24) | <0.001 | | |
| Poorest | 1.86 (1.45,2.38) | <0.001 | | |
| **Type of residence** | | | | |
| Urban | 1.00 | | | |
| Rural | 1.36 (1.14,1.63) | <0.001 | | |
| **Number of children under 5 years** | | | | |
| None | 1.00 | | | |
| 1 to 2 | 1.07 (0.66,1.72) | 0.780 | | |
| ≥3 | 2.04 (1.19,3.49) | 0.010 | | |
| **Household Size (grouped)** | | | | |
| 1 to 5 | 1.00 | | | |
| >6 | 1.25 (1.09,1.42) | <0.001 | | |
| **Source of drinking water** | | | | |
| Not improved | 1.00 | | | |
| Improved | 0.83 (0.71,0.98) | 0.030 | | |
| **Type of toilet facility** | | | | |
| Improved | 1.00 | | | |
| Unimproved | 1.71 (1.41,2.08) | <0.001 | | |
| *Maternal Factors* | | | | |
| **Respondent worked in the last 12 months** | | | | |
| Non-working | 1.00 | | | |
| Working | 1.09 (0.91,1.31) | 0.330 | | |
| **Mother's Education** | | | | |
| Secondary | 1.00 | | 1.00 | |
| Primary | 1.37 (1.10,1.69) | <0.001 | 1.10 (0.86,1.39) | 0.430 |
| No education | 2.29 (1.79,2.93) | <0.001 | 1.54 (1.18,2.00) | <0.001 |
| **Mother's age (years)** | | | | |
| 15–19 | 1.00 | | | |
| 20–34 | 1.12 (0.72,1.74) | 0.610 | | |
| 35–49 | 1.05 (0.69,1.60) | 0.820 | | |
| **Mother's marital status** | | | | |
| Never in marriage | 1.00 | | | |
| Currently married | 1.14 (0.92,1.41) | 0.230 | | |
| **Maternal anaemia** | | | | |
| Non-anaemic | 1.00 | | 1.00 | |
| Anaemic | 1.87 (1.64,2.14) | <0.001 | 1.77 (1.54,2.04) | <0.001 |
| **Maternal BMI (Kg/m$^2$)** | | | | |
| 25+ | 1.00 | | | |
| 18.5–24.9 | 1.54 (1.31,1.82) | <0.001 | | |
| <18.5 | 1.38 (0.99,1.92) | 0.060 | | |
| **Currently Pregnant** | | | | |

*(Continued)*

**Table 4.** (Continued)

| Characteristics | Unadjusted | | Adjusted | |
|---|---|---|---|---|
| | OR (95% CI) | P-Value | OR (95% CI) | P-Value |
| No | 1.00 | | | |
| Yes | 1.19 (0.95,1.50) | 0.120 | | |
| **Taking iron during pregnancy** | | | | |
| No | 1.00 | | | |
| Yes | 0.79 (0.63,0.99) | 0.040 | | |
| **Combined mode and place of delivery** | | | | |
| Vaginal | 1.00 | | 1.00 | |
| Caesarean section | 0.53 (0.37,0.75) | <0.001 | 0.55 (0.38,0.80) | <0.001 |
| Home | 1.52 (1.29,1.78) | <0.001 | 1.27 (1.09,1.48) | <0.001 |
| **Delivery assistance** | | | | |
| Health professional | 1.00 | | | |
| Non-health professional | 1.60 (1.39,1.85) | <0.001 | | |
| **Antenatal Clinical visits** | | | | |
| 8+ | 1.00 | | | |
| 4 to 7 | 1.59 (0.75,3.37) | 0.230 | | |
| 1 to 3 | 1.66 (0.79,3.47) | 0.181 | | |
| None | 1.74 (0.84,3.63) | 0.137 | | |
| **Mother read magazine or newspaper** | | | | |
| At least | 1.00 | | | |
| Less than | 1.06 (0.88,1.27) | 0.520 | | |
| Never | 1.11 (0.95,1.30) | 0.180 | | |
| **Mother listened to radio** | | | | |
| At least | 1.00 | | | |
| Less than | 1.29 (1.09,1.53) | <0.001 | | |
| Never | 1.42 (1.19,1.71) | <0.001 | | |
| **Mother watched television** | | | | |
| At least | 1.00 | | | |
| Less than | 1.39 (1.11,1.73) | <0.001 | | |
| Never | 1.60 (1.31,1.96) | <0.001 | | |
| **Mother power over earnings** | | | | |
| No | 1.00 | | | |
| Yes | 0.83 (0.72,0.96) | 0.010 | | |
| **Mother power over household decision making** | | | | |
| No | 1.00 | | | |
| Yes | 0.87 (0.75,1.00) | 0.050 | | |
| **Mother autonomy over health care** | | | | |
| No | 1.00 | | | |
| Yes | 0.82 (0.71,0.94) | 0.010 | | |
| *Child Factors* | | | | |
| **Combined birth interval and birth order** | | | | |
| 1st birth rank | 1.00 | | | |
| 2nd/3rd birth rank, more than 2 years interval | 0.96 (0.80,1.16) | 0.720 | | |
| 2nd/3rd birth rank, less than or equal to 2 | 1.12 (0.86,1.47) | 0.370 | | |
| 4th birth rank, more than 2 years interval | 1.19 (0.98,1.44) | 0.070 | | |
| 4th birth rank, less than or equal to 2 | 1.26 (0.94,1.67) | 0.110 | | |
| **Sex of child** | | | | |

*(Continued)*

**Table 4.** (Continued)

| Characteristics | Unadjusted | | Adjusted | |
|---|---|---|---|---|
| | OR (95% CI) | P-Value | OR (95% CI) | P-Value |
| Female | 1.00 | | | |
| Male | 1.07 (0.94,1.23) | 0.280 | | |
| **Size of baby** | | | | |
| Large | 1.00 | | | |
| Average | 0.91 (0.77,1.09) | 0.340 | | |
| Small | 0.95 (0.72,1.25) | 0.720 | | |
| **Stunted** | | | | |
| No | 1.00 | | 1.00 | |
| Yes | 1.60 (1.38,1.85) | <0.001 | 1.46 (1.25,1.71) | <0.001 |
| **Wasted** | | | | |
| No | 1.00 | | | |
| Yes | 1.06 (0.70,1.60) | 0.780 | | |
| **Underweight** | | | | |
| No | 1.00 | | | |
| Yes | 1.25 (1.01,1.56) | 0.040 | | |
| **Children <5 slept with bed net** | | | | |
| Yes | 1.00 | | | |
| No | 0.90 (0.88, 1.08) | 0.210 | | |
| **Given drugs for preventing worm infestation** | | | | |
| No | 1.00 | | 1.00 | |
| Yes | 0.73 (0.63,0.84) | <0.001 | 0.84 (0.72,0.98) | 0.030 |
| **Fully Vaccinated** | | | | |
| No | 1.00 | | | |
| Yes | 1.41 (0.76,2.60) | 0.270 | | |
| **Vitamin A** | | | | |
| No | 1.00 | | | |
| Yes | 0.84 (0.73,0.96) | 0.020 | | |
| **Had diarrhoea in the last two weeks** | | | | |
| No | 1.00 | | | |
| Yes | 1.11 (0.86,1.41) | 0.400 | | |
| **Had fever in the last two weeks** | | | | |
| No | 1.00 | | 1.00 | |
| Yes | 1.64 (1.35,1.99) | <0.001 | 1.61 (1.32,1.95) | <0.001 |

In this study, we found that none or low level of maternal education was associated with iron deficiency anaemia in children. Surprisingly, we found that maternal employment was associated with iron deficiency anaemia in children, contrary to findings from previous research from Brazil [24, 25]. Unemployment and low level of education may lead to poor socio-economic status, thus suggesting that better socio-demographic conditions may increase access to better nutrition and health care and, as a consequence, lower the risk of anaemia [13]. Our finding of the association between a low level of maternal education and iron deficiency anaemia is consistent with a past study from Tanzania [13]. A previous study from Tanzania observed that anaemic children had mothers/carers who did not complete primary education [26]. Another previous study from Ethiopia [27] revealed that the education of mothers demonstrated a protective effect on childhood anaemia. This could be explained by the use of improved feeding and childcare practice by educated mothers. This Ethiopian study

**Table 5. Univariate and multivariate analyses of the odds of a child aged 6–59 months being anaemic–Tanzania 2015–16.**

| Characteristics | Unadjusted | | Adjusted | |
|---|---|---|---|---|
| | OR (95% CI) | P-Value | OR (95% CI) | P-Value |
| *Household Factors* | | | | |
| **Household Wealth Index** | | | | |
| Richest | 1 | | | |
| Richer | 1.02 (0.85,1.23) | 0.821 | | |
| Middle | 1.36 (1.12,1.64) | 0.002 | | |
| Poorer | 1.44 (1.20,1.74) | <0.001 | | |
| Poorest | 1.60 (1.33,1.91) | <0.001 | | |
| **Type of place of residence** | | | | |
| Urban | 1 | | | |
| Rural | 1.23 (1.06,1.41) | 0.005 | | |
| **Household Size (grouped)** | | | | |
| 1 to 5 | 1 | | 1 | |
| >6 | 1.22 (1.09,1.36) | <0.001 | 1.15 (1.02,1.30) | 0.022 |
| **Source of drinking water** | | | | |
| Not improved | 1 | | | |
| Improved | 0.84 (0.74,0.96) | 0.013 | | |
| **Type of toilet facility** | | | | |
| Improved | 1 | | | |
| Unimproved | 1.50 (1.29,1.75) | <0.001 | | |
| *Maternal Factors* | | | | |
| **Respondent worked in the last 12 months** | | | | |
| Non-working | 1 | | 1 | |
| Working | 1.60 (1.33,1.91) | 0.001 | 1.20 (1.03,1.38) | 0.013 |
| **Mother's Education** | | | | |
| Secondary | 1 | | 1 | |
| Primary | 1.08 (0.92,1.26) | 0.372 | 1.05 (0.87,1.27) | 0.548 |
| No education | 1.59 (1.31,1.94) | <0.001 | 1.46 (1.17,1.84) | 0.001 |
| **Mother's age (years)** | | | | |
| 15–19 | 1 | | | |
| 20–34 | 1.60 (1.33,1.91) | <0.001 | | |
| 35–49 | 1.60 (1.33,1.91) | <0.001 | | |
| **Mother's marital status** | | | | |
| Never in marriage | 1 | | | |
| Currently married | 1.07 (0.89,1.29) | 0.462 | | |
| **Maternal anaemia** | | | | |
| Non-anaemic | 1 | | 1 | |
| Anaemic | 1.81 (1.63,2.01) | <0.001 | 1.86 (1.65,2.09) | <0.001 |
| **Maternal BMI (Kg/m$^2$)** | | | | |
| 25+ | 1 | | 1 | |
| 18.5–24.9 | 1.60 (1.33,1.91) | <0.001 | 1.25 (1.08,1.45) | 0.002 |

(*Continued*)

**Table 5.** (Continued)

| Characteristics | Unadjusted | | Adjusted | |
|---|---|---|---|---|
| | OR (95% CI) | P-Value | OR (95% CI) | P-Value |
| <18.5 | 1.60 (1.33,1.91) | 0.001 | 1.12 (0.82,1.51) | 0.453 |
| **Currently Pregnant** | | | | |
| No | 1 | | | |
| Yes | 1.00 (0.81,1.22) | 0.980 | | |
| **Taking iron during pregnancy** | | | | |
| No | 1 | | | |
| Yes | 0.94 (0.79,1.11) | 0.460 | | |
| **Combined mode and place of delivery** | | | | |
| Vaginal | 1 | | 1 | |
| Caesarean section | 0.69 (0.53,0.89) | 0.004 | 0.65 (0.50,0.85) | 0.002 |
| Home | 1.27 (1.12,1.45) | <0.001 | 1.15 (1.01,1.32) | 0.038 |
| **Delivery assistance** | | | | |
| Health professional | 1 | | | |
| Non-health professional | 1.31 (1.16,1.48) | <0.001 | | |
| **Antenatal Clinical visits** | | | | |
| 8+ | 1 | | | |
| 4 to 7 | 1.72 (0.94,3.12) | 0.076 | | |
| 1 to 3 | 1.91 (1.06,3.44) | 0.031 | | |
| None | 0.78 (0.65,2.12) | <0.001 | | |
| **Mother read magazine or newspaper** | | | | |
| At least | 1 | | | |
| Less than | 0.94 (0.76,1.17) | 0.589 | | |
| Never | 1.12 (0.93,1.36) | 0.245 | | |
| **Mother listened to radio** | | | | |
| At least | 1 | | | |
| Less than | 1.17 (1.02,1.33) | 0.022 | | |
| Never | 1.28 (1.11,1.47) | 0.001 | | |
| **Mother watched television** | | | | |
| At least | 1 | | | |
| Less than | 1.29 (1.09,1.53) | 0.004 | | |
| Never | 1.32 (1.14,1.53) | <0.001 | | |
| **Mother power over earnings** | | | | |
| No | 1 | | | |
| Yes | 0.84 (0.74,0.94) | 0.002 | | |
| **Mother power over household decision making** | | | | |
| No | 1 | | | |
| Yes | 0.84 (0.75,0.94) | 0.003 | | |

(*Continued*)

**Table 5.** (Continued)

| Characteristics | Unadjusted | | Adjusted | |
|---|---|---|---|---|
| | OR (95% CI) | P-Value | OR (95% CI) | P-Value |
| **Mother autonomy over health care** | | | | |
| No | 1 | | | |
| Yes | 0.81 (0.73,0.91) | <0.001 | | |
| *Child factors* | | | | |
| **Combined birth interval and birth order** | | | | |
| 1st birth rank | 1 | | | |
| 2nd/3rd birth rank, more than 2 years interval | 0.84 (0.72,0.97) | 0.019 | | |
| 2nd/3rd birth rank, less than or equal to 2 | 0.96 (0.78,1.18) | 0.723 | | |
| 4th birth rank, more than 2 years interval | 0.98 (0.84,1.14) | 0.822 | | |
| 4th birth rank, less than or equal to 2 | 0.94 (0.75,1.17) | 0.590 | | |
| **Sex of child** | | | | |
| Female | 1 | | 1 | |
| Male | 1.14 (1.03,1.26) | 0.010 | 1.14 (1.02,1.27) | 0.021 |
| **Size of baby** | | | | |
| Large | 1 | | | |
| Average | 1.01 (0.88,1.16) | 0.785 | | |
| Small | 1.17 (0.95,1.46) | 0.133 | | |
| **Child's age in Months** | | | | |
| 6 to 23 | 1 | | 1 | |
| 24 to 59 | 3.31 (2.97,3.68) | <0.001 | 3.66 (3.27,4.10) | <0.001 |
| **Stunted** | | | | |
| No | 1 | | 1 | |
| Yes | 1.26 (1.12,1.42) | <0.001 | 1.31 (1.14,1.50) | <0.001 |
| **Wasted** | | | | |
| No | 1 | | | |
| Yes | 1.48 (1.13,1.93) | 0.004 | | |
| **Underweight** | | | | |
| No | 1 | | | |
| Yes | 1.22 (1.03,1.45) | 0.018 | | |
| **Children <5 slept with bed net** | | | | |
| Yes | 1 | | | |
| No | 0.90 (0.79,1.03) | 0.163 | | |
| **Given drugs for preventing worm infestation** | | | | |
| No | 1 | | | |
| Yes | 0.67 (0.59,0.75) | <0.001 | | |
| **Number of children under 5 years** | | | | |

(*Continued*)

**Table 5.** (Continued)

| Characteristics | Unadjusted | | Adjusted | |
|---|---|---|---|---|
| | OR (95% CI) | P-Value | OR (95% CI) | P-Value |
| None | 1 | | | |
| 1 to 2 | 0.98 (0.63,1.53) | 0.930 | | |
| ≥3 | 1.75 (1.08,2.85) | 0.024 | | |
| **Fully Vaccinated** | | | | |
| No | 1 | | | |
| Yes | 0.99 (0.56,1.74) | 0.990 | | |
| **Vitamin A** | | | | |
| No | 1 | | | |
| Yes | 0.87 (0.77,0.97) | 0.020 | | |
| **Had diarrhoea in the last two weeks** | | | | |
| No | 1 | | | |
| Yes | 1.56 (1.32,1.85) | <0.001 | | |
| **Had fever in the last two weeks** | | | | |
| No | 1 | | 1 | |
| Yes | 1.55 (1.34,1.79) | <0.001 | 1.44 (1.23,1.68) | <0.001 |

& = doctor; nurse/midwife/paramedics, family welfare visitor medical assistant/community medical officer/ health assistant

also observed that the children of mothers with only primary education had comparable levels of anaemia to their counterparts whose mothers were had secondary education or higher. The level of education of mothers has been previously shown to decrease the prevalence of anaemia [28, 29]. The association of childhood iron deficiency anaemia and education may be attributed to the ability of mothers/carers to acquire the required knowledge for adequate health care and nutrition for children.

We found that stunting was significantly associated with iron deficiency anaemia among children aged 24–59 months and 6–59 months (unlike the other nutrition indices of wasting and underweight), in consonance with previous studies from Brazil [30, 31], Burma [32] and Ethiopia [33]. This association may be due to the fact that i) undernourished children are often anaemic, ii) low haemoglobin concentration can compromise linear growth, and iii) the coexistence of other micronutrient deficiencies and stunting may increase the development of anaemia by a synergistic association. Consequently, efforts geared towards preventing stunting in children may also contribute to reducing the iron deficiency anaemia menace in children.

Small-sized children (low birth weight) aged 6–23 months were found to be associated with anaemia in this study. This finding is corroborated by previous findings from Tanzania [34] and Malawi [35]. Although low birth weight has been strongly associated with maternal iron deficiency anaemia [36, 37], which negatively impacts a child's iron store at birth, a clear association between baby size and anaemia has not been established.

Our current study revealed an association between childhood iron deficiency anaemia and low socio-economic status. In spite of a dearth of data in the extant literature which identifies the risk factors for iron deficiency anaemia in children aged less than two years, past studies have observed that socio-economic and environmental factors appear to be closely associated with iron deficiency anaemia among these children [32, 33, 38]. Children with mothers from

poor households had a statistically significant increased level of childhood anaemia. This finding has also been demonstrated in similar studies [28, 29].

In this study, we found a significant association between anaemic mothers and childhood iron deficiency anaemia. This finding is consistent with other past studies [39]. In addition, there is evidence of an association of maternal iron deficiency during pregnancy and lactation and iron deficiency in young children [39]. Furthermore, a study conducted in India [39] revealed that severe maternal iron deficiency anaemia can adversely affect cord blood and breast milk iron status, which may lead to serious consequences for the young infant at a time when demands for iron are high. Furthermore, evidence from the extant literature suggests that maternal iron deficiency may compromise maternal cognitive and interactive behaviours, which could influence the child's care [40]. Consequently, it is important to ensure adequate maternal nutritional status during this important period which favours adequate development of young children.

Our study revealed that being a male child was significantly associated with increased odds of childhood iron deficiency anaemia. This finding is consistent with previous studies from Kenya [41, 42]. A similar finding was made in another study from Ghana [43]. Greater attention should be given to male children in terms of dietary diversity and iron-rich complementary foods.

Evidence in this study that was mentioned above, the prevalence suggests the occurrence of iron deficiency anaemia among children aged 6 to 59 months as severe public health, in turn associated with either proximal or distal determinants. The nutrition-specific approach mainly targeting the immediate/proximal determinants and, can be delivered through education, communication, social marketing, and behaviour changing programs to target specific groups, and in young children, they are frequently oriented to improve age adequate breastfeeding, dietary diversity, and feeding frequency [2, 44]. Conversely, a nutrition-sensitive approach is targeting fundamental/distal determinants from a wide range of sectors, comprising disease control, water, sanitation, and hygiene and inter-sectoral strategies that address root causes (e.g., poverty, lack of education, and gender norms) [2, 44]. Therefore guiding future strategies targeting nutritional anaemia among children aged 6 to 59 months in Tanzania should be combined; Integrating strategies on preventive (either nutrition-based or WASH/ malaria education) and therapeutic (deworming) could result in the reduction of anaemia [2, 44]. For instance, deworming plus nutrition education has been reported to reduce anaemia from 82.0% to 55.4%, while increasing green leafy vegetable consumption from 44.7% to 60.6%.40 Furthermore, exclusive educational interventions targeting nutrition improvement; the increased ability of mothers to identify malnutrition (from 15% to 99%); exclusive breastfeeding (79% compared to 48% in control); weight gain (0.86 compared to 0.77 kg in control); vegetables feeding; nutrient-dense foods at lunch (11% of the difference between intervention and control groups); dietary requirements for energy, iron, and zinc; complementary feeding; and significantly reduced rates of stunting [45–47].

One of the important strengths of this study is that the 2015–16 TDHS is nationally representative and used a standardised method to achieve a high response rate, which meant that the findings in the study would be devoid of sampling bias in Tanzania. However, the study is not without limitations, which should be considered when interpreting the results of the study. Firstly, the cross-sectional nature of this study is a potential limitation because this kind of study does not allow inference of causality, and results must thus be interpreted carefully. Secondly, despite the WHO recommendation in 2014 on delayed umbilical cord clamping and cutting to reduce the risk of anaemia and hypoxia in infants, however, its implementation is still low in many countries including Tanzania [48, 49]. Thirdly, haemolytic anaemia attributed by thalassemia and other haemoglobinopathies was not part of this study as it was not assessed in the 2015–16 TDHS. Although the occurrence of haemoglobinopathies in Tanzania

fulfils the World Health Organization criteria of public health importance, this problem is often overlooked. For instance, no country in sub- Saharan Africa has even established a national universal newborn screening (NBS) programme [11] and lastly, this study used MDD-7, meaning the proportion of children aged 6–23 months who consumed $\geq$ 4 of 7 food groups instead of $\geq$ 5 of 8 food groups (MDD-8) as reported in [50]. However, a recent study conducted in the Eastern and Southern Africa region by Heidkamp et al. [51] that compared the prevalence of MDD-8 and MDD-7 revealed that confidence Intervals around the prevalence estimates for MDD-7 [OR25.6%; 95%CI: 23.4, 27.9] and MDD-8 [OR 20.2%; 95%CI: 18.7, 22.7] overlapped for Tanzania [51].

## Conclusions

Childhood anaemia in Tanzania is a significant public health problem. A higher degree of maternal anaemia predisposes to childhood iron deficiency anaemia. Low maternal level of education and low household economic status were found to be associated with iron deficiency anaemia in children. Identifying the root cause of iron deficiency anaemia in children and improving nutritional advice and supplementation will be required. Prospective cohort studies on haemoglobinopathies contribution to be burden of childhood anaemia may be an important field for future research. It is important to develop public health interventions with a holistic approach to mothers and children. Efforts to deliver the Sustainable Development Goals meant to improve education for women and eradicate poverty also have the likelihood of leading to significant reductions in childhood iron deficiency anaemia.

## Supporting information

**S1 Table. Definition and categorisation of potential variables used in the study.** (DOCX)

## Author Contributions

**Conceptualization:** Rose V. Msaki, Elizabeth Lyimo, Ray M. Masumo, Eliasaph Mwana, Doris Katana, Adeline Munuo, Germana Leyna, Abukari I. Issaka, Kingsley E. Agho.

**Data curation:** Rose V. Msaki.

**Formal analysis:** Rose V. Msaki, Elizabeth Lyimo, Ray M. Masumo, Eliasaph Mwana, Germana Leyna, Abukari I. Issaka, Mansi V. Dhami, Kingsley E. Agho.

**Methodology:** Rose V. Msaki, Elizabeth Lyimo, Ray M. Masumo, Eliasaph Mwana, Germana Leyna, Abukari I. Issaka, Mansi V. Dhami, Kingsley E. Agho.

**Project administration:** Rose V. Msaki, Elizabeth Lyimo, Germana Leyna.

**Software:** Germana Leyna.

**Supervision:** Ray M. Masumo, Germana Leyna, Kingsley E. Agho.

**Validation:** Rose V. Msaki.

**Writing – original draft:** Rose V. Msaki, Elizabeth Lyimo, Ray M. Masumo, Eliasaph Mwana, Doris Katana, Nyamizi Julius, Adeline Munuo, Abukari I. Issaka, Mansi V. Dhami, Kingsley E. Agho.

**Writing – review & editing:** Rose V. Msaki, Elizabeth Lyimo, Eliasaph Mwana, Germana Leyna, Abukari I. Issaka, Mansi V. Dhami, Kingsley E. Agho.

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
