## [Decision Letter · Decision Letter 0]

23 Sep 2021

 PGPH-D-21-00225 Predictors of anaemia among children aged 6–59 months in Tanzania: evidence from the 2015-16 TDHS-MIS cross-sectional household survey PLOS Global Public Health

Dear Dr. Masumo,

Thank you for submitting your manuscript to PLOS Global Public Health. After careful consideration, we feel that it has merit but does not fully meet PLOS Global Public Health’s publication criteria as it currently stands. Therefore, we invite you to submit a revised version of the manuscript that addresses the points raised during the review process.

Major Revisions

We look forward to receiving your revised manuscript.

Kind regards,

Faisal Abbas

Academic Editor

Journal Requirements:

1. Please amend your detailed Financial Disclosure statement. This is published with the article, therefore should be completed in full sentences and contain the exact wording you wish to be published.

2. In the online submission form, you indicated that "All datasets are available upon request from the DHS website (https://dhsprogram.com/)". All PLOS journals now require all data underlying the findings described in their manuscript to be freely available to other researchers, either 1. In a public repository, 2. Within the manuscript itself, or 3. Uploaded as supplementary information.

3. Please provide separate figure files in .tif or .eps format only and remove any figures embedded in your manuscript file. Please ensure that all files are under our size limit of 20MB.  

Once you've converted your files to .tif or .eps, please also make sure that your figures meet our format requirements

4. Please ensure that you refer to Figure 2 in your text as, if accepted, production will need this reference to link the reader to the figure.

5. We notice that your supplementary tables are included in the manuscript file. Please remove them and upload them with the file type 'Supporting Information'. Please ensure that all Supporting Information files are included correctly and that each one has a legend listed in the manuscript after the references list.

Additional Editor Comments (if provided):

Major Revisions are required.

Reviewers' comments:

Reviewer's Responses to Questions

**Comments to the Author**

1. Does this manuscript meet PLOS Global Public Health’s publication criteria? Is the manuscript technically sound, and do the data support the conclusions? The manuscript must describe methodologically and ethically rigorous research with conclusions that are appropriately drawn based on the data presented.

Reviewer #1: Partly

Reviewer #2: Partly

2. Has the statistical analysis been performed appropriately and rigorously?

Reviewer #1: Yes

Reviewer #2: Yes

3. Have the authors made all data underlying the findings in their manuscript fully available (please refer to the Data Availability Statement at the start of the manuscript PDF file)?

Reviewer #1: Yes

Reviewer #2: No

4. Is the manuscript presented in an intelligible fashion and written in standard English?

Reviewer #1: Yes

Reviewer #2: Yes

5. Review Comments to the Author

Reviewer #1: No doubt the paper is of public health importance. But, there must be some mention and findings regarding:

1. Feeding pattern of infants, especially whether breast-fed or not.

2. Exclusion of family history of Thalassemia and sickle-cell anemia.

3. Inclusion and exclusion criteria of study population.

Reviewer #2: this study intended to assess factors associated with anaemia among children aged 6-59 months and analysed maternal, child and household various factors which all related to iron deficiency anaemia. But to find out the actual picture of anaemia, it is very important to rule out thalassemia, sickle cell disease, haemoglobinopathies and carrier status of mother and child. So, the study data is not sufficient to support the conclusions.

Supplied link does not specify the data underlying the findings described in the manuscript.

6. PLOS authors have the option to publish the peer review history of their article (what does this mean?). If published, this will include your full peer review and any attached files.

**Do you want your identity to be public for this peer review?** For information about this choice, including consent withdrawal, please see our Privacy Policy.

Reviewer #1: **Yes: **Prof. (Dr.) Manzoor Hussain

Reviewer #2: No

---

## [Decision Letter · Decision Letter 1]

17 Mar 2022

PGPH-D-21-00225R1

Predictors of anaemia among children aged 6–59 months in Tanzania: evidence from the 2015-16 TDHS-MIS cross-sectional household survey

Dear Dr. Masumo,

Thank you for submitting your manuscript to PLOS Global Public Health. After careful consideration, we feel that it has merit but does not fully meet PLOS Global Public Health’s publication criteria as it currently stands. Therefore, we invite you to submit a revised version of the manuscript that addresses the points raised during the review process.

Major Revisions

We look forward to receiving your revised manuscript.

Kind regards,

Faisal Abbas, PhD

Academic Editor

Journal Requirements:

Additional Editor Comments (if provided):

Major Revisions

Reviewers' comments:

Reviewer's Responses to Questions

**Comments to the Author**

1. If the authors have adequately addressed your comments raised in a previous round of review and you feel that this manuscript is now acceptable for publication, you may indicate that here to bypass the “Comments to the Author” section, enter your conflict of interest statement in the “Confidential to Editor” section, and submit your "Accept" recommendation.

Reviewer #1: All comments have been addressed

Reviewer #2: (No Response)

2. Does this manuscript meet PLOS Global Public Health’s publication criteria? Is the manuscript technically sound, and do the data support the conclusions? The manuscript must describe methodologically and ethically rigorous research with conclusions that are appropriately drawn based on the data presented.

Reviewer #1: Yes

Reviewer #2: Partly

3. Has the statistical analysis been performed appropriately and rigorously?

Reviewer #1: I don't know

Reviewer #2: Yes

4. Have the authors made all data underlying the findings in their manuscript fully available (please refer to the Data Availability Statement at the start of the manuscript PDF file)?

Reviewer #1: Yes

Reviewer #2: Yes

5. Is the manuscript presented in an intelligible fashion and written in standard English?

Reviewer #1: Yes

Reviewer #2: Yes

6. Review Comments to the Author

Reviewer #1: May be published.

Reviewer #2: Authors totally ignored the comments raised in previous round of review. Hemolytic anemia such as thalassemia and other hemoglobinopathies must be find out or excluded to evaluate the early childhood anaemia and the risk factors. So, I feel that, this manuscript is not acceptable for publication.

7. PLOS authors have the option to publish the peer review history of their article (what does this mean?). If published, this will include your full peer review and any attached files.

**Do you want your identity to be public for this peer review?** For information about this choice, including consent withdrawal, please see our Privacy Policy.

Reviewer #1: **Yes: **Manzoor Hussain

Reviewer #2: No

---

## [Decision Letter · Decision Letter 2]

29 Jul 2022

PGPH-D-21-00225R2

Predictors of anaemia among children aged 6–59 months in Tanzania: evidence from the 2015-16 TDHS-MIS cross-sectional household survey

Dear Dr. Masumo,

Thank you for submitting your manuscript to PLOS Global Public Health. After careful consideration, we feel that it has merit but does not fully meet PLOS Global Public Health’s publication criteria as it currently stands. Therefore, we invite you to submit a revised version of the manuscript that addresses the points raised during the review process.

We look forward to receiving your revised manuscript.

Kind regards,

Marianella Herrera-Cuenca, MD, PhD

Academic Editor

Journal Requirements:

Additional Editor Comments (if provided):

Reviewers' comments:

Reviewer's Responses to Questions

**Comments to the Author**

1. If the authors have adequately addressed your comments raised in a previous round of review and you feel that this manuscript is now acceptable for publication, you may indicate that here to bypass the “Comments to the Author” section, enter your conflict of interest statement in the “Confidential to Editor” section, and submit your "Accept" recommendation.

Reviewer #3: (No Response)

2. Does this manuscript meet PLOS Global Public Health’s publication criteria? Is the manuscript technically sound, and do the data support the conclusions? The manuscript must describe methodologically and ethically rigorous research with conclusions that are appropriately drawn based on the data presented.

Reviewer #3: Yes

3. Has the statistical analysis been performed appropriately and rigorously?

Reviewer #3: Yes

4. Have the authors made all data underlying the findings in their manuscript fully available (please refer to the Data Availability Statement at the start of the manuscript PDF file)?

Reviewer #3: Yes

5. Is the manuscript presented in an intelligible fashion and written in standard English?

Reviewer #3: Yes

6. Review Comments to the Author

Reviewer #3: Is umbilical cord ligation after 30 seconds to 3 minutes a universal practice? Recognizing its impact on iron stores early in childhood, it can bias the result in the population studied

7. PLOS authors have the option to publish the peer review history of their article (what does this mean?). If published, this will include your full peer review and any attached files.

**Do you want your identity to be public for this peer review?** For information about this choice, including consent withdrawal, please see our Privacy Policy.

Reviewer #3: **Yes: **Maria Jose Castro

---

## [Editor Report · Decision Letter 3]

26 Aug 2022

PGPH-D-21-00225R3

Predictors of anaemia among children aged 6–59 months in Tanzania: evidence from the 2015-16 TDHS-MIS cross-sectional household survey

Dear Dr. Masumo,

Thank you for submitting your manuscript to PLOS Global Public Health. After careful consideration, we feel that it has merit but does not fully meet PLOS Global Public Health’s publication criteria as it currently stands. Therefore, we invite you to submit a revised version of the manuscript that addresses the points raised during the review process.

We look forward to receiving your revised manuscript.

Kind regards,

Marianella Herrera-Cuenca, MD, PhD

Academic Editor

Journal Requirements:

Additional Editor Comments (if provided):

This is a very important topic for public health however you need to consider some major revisions such as the title of the article: did you addressed all micronutrients related to anemia? no, then why dont you make some more specific duch as ferropenic anemia (just an idea) please check carefully all revisors comments and work on them
---

## [Editor Report · Decision Letter 4]

14 Oct 2022

Predictors of iron deficiency anaemia among children aged 6–59 months in Tanzania: evidence from the 2015-16 TDHS-MIS cross-sectional household survey

PGPH-D-21-00225R4

Dear Dr Masumo,

We are pleased to inform you that your manuscript 'Predictors of iron deficiency anaemia among children aged 6–59 months in Tanzania: evidence from the 2015-16 TDHS-MIS cross-sectional household survey' has been provisionally accepted for publication in PLOS Global Public Health.

Best regards,

Marianella Herrera-Cuenca, MD, PhD

Academic Editor